# Antidepressant Potential of *Lotus corniculatus* L. subsp. *corniculatus*: An Ethnobotany Based Approach

**DOI:** 10.3390/molecules25061299

**Published:** 2020-03-12

**Authors:** Fatma Tuğçe Gürağaç Dereli, Haroon Khan, Eduardo Sobarzo-Sánchez, Esra Küpeli Akkol

**Affiliations:** 1Department of Pharmacognosy, Faculty of Pharmacy, Gazi University, 06330 Etiler, Ankara, Turkey; ecztugceguragac@gmail.com; 2Department of Pharmacy, Abdul Wali Khan University, Mardan 23200, Pakistan; haroonkhan@awkum.edu.pk; 3Instituto de Investigación e Innovación en Salud, Facultad de Ciencias de la Salud, Universidad Central de Chile, 8330507 Santiago, Chile; 4Department of Organic Chemistry, Faculty of Pharmacy, University of Santiago de Compostela, 15782 Santiago de Compostela, Spain

**Keywords:** antidepressant, *Lotus corniculatus*, Fabaceae, forced swimming test, monoamine oxidase, tail suspension test, tetrabenazine

## Abstract

As a Turkish traditional medicinal plant, aerial parts of *Lotus corniculatus* L. subsp. *corniculatus* (Fabaceae) are used as a painkiller, antihemoroidal, diuretic and sedative. In this study, the antidepressant potential of the plant has been attempted to clarify. Extracts with water, *n*-Hexane, ethyl acetate, and methanol were prepared respectively from the aerial parts. Antidepressant activity of the extracts were researched by using three different in vivo test models namely a tail suspension test, antagonism of tetrabenazine-induced hypothermia, ptosis, and suppression of locomotor activity and forced swimming test on male BALB/c mice and in vitro monoamine oxidase (MAO)-A and B inhibition assays. The results were evaluated through comparing with control and reference groups, and then active compounds of the active extract have been determined. Bioassay-guided fractionation of active fraction led to the isolation of three compounds and structures of the compounds were elucidated by spectroscopic methods. The data of this study demonstrate that the MeOH extract of the aerial parts of the plant showed remarkable in vivo antidepressant effect and the isolated compounds medicarpin-3-*O*-glucoside, gossypetin-3-*O*-glucoside and naringenin-7-*O*-glucoside (prunin) from the active sub-fractions could be responsible for the activity. Further mechanistic and toxicity studies are planned to develop new antidepressant-acting drugs.

## 1. Introduction

Depression is a common but serious mood disorder and its etiology has not yet been elucidated. The studies to date have shown that the onset of this disease may be associated with several genetic, biological and psychosocial risk factors [1]. The symptoms of depression can vary from mild to severe and include emotional, cognitive, behavioral and physical changes and all of these can cause disruptions in normal daily activities of individuals [2,3]. In addition to psychosocial problems, it can lead to some other additional diseases such as asthma, diabetes, obesity and cancer [4]. The World Health Organization estimate that over 300 million people suffer from depression and about 800,000 people die each year in suicide cases due to it [5]. The treatment of this serious public health problem is very important because of its financial and moral damages. 

The three most known treatment options for depression are psychotherapy, antidepressant medication and brain stimulation therapies. However, the most preferred treatment is a combination of the medication and cognitive behavioral therapy. There are many different types of antidepressants that can help reduce the symptoms of depression including monoamine oxidase (MAO) inhibitors, tricyclic antidepressants, specific serotonin-noradrenaline reuptake inhibitors and atypical antidepressants [6]. Due to the common serious side effects of available depression medicines such as weight gain, insomnia and sexual dysfunction, researchers continue to search for new alternative drug molecules in the treatment [7]. The ethnobotanical-based approach is particularly important for diseases such as depression that have not achieved a satisfactory therapeutic level. In such studies, antidepressant activities of various phytochemicals have been tried to be elucidate in the light of ethnobotanical information [8].

In this current study, the antidepressant activity of *Lotus corniculatus* L. subsp. *corniculatus* from Fabaceae family has been attempted to clarify. For aught we know, no report about the antidepressant potential of the plant is available to date. As a Turkish traditional medicinal plant, aerial parts of the plant are used as painkiller, antihemorrhoidal, diuretic and sedative [9]. The bioassay-guided fractionation process was preferred to investigation of the main active chemical compounds of the plant. 

## 2. Results

### 2.1. Biological Activity Studies

In this study, in vivo antidepressant activity of a Turkish medicinal plant, *L. corniculatus* subsp. *corniculatus* was investigated by using three different in vivo test models namely forced swimming test, tail suspension test and antagonism of tetrabenazine-induced ptosis, hypothermia and suppression of locomotor activity and in vitro MAO inhibition assay. 

Firstly, the aqueous extract was prepared according to traditional usage. In vivo tests results for aqueous extract were summarized in Table 1. As shown, the aqueous extract did not show any remarkable antidepressant activity. Due to this extract did not show any remarkable antidepressant activity, the activity of three different extracts was obtained by successive extraction with n-hexane, EtOAc and MeOH were evaluated. 

The MeOH extract reduced the immobility period by 33.40% (*p* < 0.05) compared to the control group and this result was found to be statistically significant (Table 2). 

As shown in Table 3, similar results were received for TST. The MeOH extract shortened the immobility time significantly with the value of 38.11% (*p* < 0.05) compared to the control group and this reduction was found to be statistically significant. 

Similar results were obtained in the antagonism of ptosis and hypothermia induced by tetrabenazine test, as indicated in Table 4. The MeOH extract increased the locomotor activity by 33.3%, reduced the ptosis score to 1.75 and changed rectal temperature with a decrease of 1.51 °C. 

However, the extracts inhibited MAO-A and MAO-B enzymes with the high IC_50_ values in the MAO inhibition assay (Table 5), all of the received results from in vivo studies led to the isolation studies on active MeOH extract. 

Reverse phase C18 (RP-18) column chromatography was performed on this extract to elute pure compounds responsible from the activity. The obtained fractions were characterized by thin-layer chromatograms and grouped into four categories (Fractions 1–42). All of the fractions inhibited MAO-A and MAO-B enzymes with the high IC_50_ values in MAO inhibition assay and of these fractions, only the Fr. (13–20) fraction showed the most potent activity in in vivo assays (Table 2, Table 3, Table 4 and Table 5). Therefore, the Fr. (13–20) fraction was fractionated to the three subfractions by Sephadex LH-20 with MeOH and three compounds were isolated.

### 2.2. Structure of Isolated Compounds

The structures of the isolates were elucidated to be medicarpin 3-*O*-glucoside, gossypetin-3-*O*-glucoside and naringenin-7-*O*-glucoside (Figure 1).

## 3. Discussion

Natural resources have been used for treatment purposes for ages and plants are the most favorite among them [10]. In this study, in vivo antidepressant activity of a Turkish medicinal plant, *L. corniculatus* subsp. *corniculatus* was investigated by using three different in vivo test models namely forced swimming test, tail suspension test, and antagonism of tetrabenazine-induced ptosis, hypothermia and suppression of locomotor activity and an in vitro MAO inhibition assay.

Phytochemical investigation studies on *L. corniculatus* reported the isolation of several flavonoid compounds and glycosides such as quercetin, naringenin, hyperoside, quercetin-*O*-deoxyhexoside-*O*-hexoside,quercetin-3-*O*-rhamnoside-7-*O*-glucoside,quercetin-3,7dirhamnoside,isoquercitrin,quercetin-*O*-pentoside,quercitrin,kaempferol-3-*O*-[xylosyl-(1→2)-galactoside]-7-*O*-rhamnoside,kaempferol-7-*O*-glucoside,afzelin,kaempferol-*O*-deoxyhexosylhexoside-*O*-deoxyhexoside, kaempferol-3-*O*-rhamnoside-7-*O*-glucoside, gossypetin, gossypetin-3-*O*-galactoside, gossytrin, medicarpin, medicarpin-3-*O*-β-d-glucopyranoside; saponoids such as soyasaponin I, dehydrosoyasaponin I, pharbitoside A; benzoic acid and cinnamic acid derivatives include p-coumaric acid, caffeic acid and chlorogenic acid so far [11,12,13,14]. However, to the best of our knowledge, there is no study about the chemical composition and also antidepressant activity of *L. corniculatus* subsp. *corniculatus*. There is at present only one bioactivity study in the literature and this one is related to the cytotoxicity potential and antioxidant effect of the plant [15]. In this sense, this study is the first in determining the activity against depression and also the chemical profile of the plant.

Plenty of studies have been designed to assess the antidepressant-like activity of flavonoids so far and activity mechanism of this type of compounds is thought to be gene regulation for reversing of monoamine neurotransmitter attenuation or neurotransmitter receptor expression [16].

Medicarpin is a pterocarpan (a derivative of isoflavonoids) and pterocarpan derivative compounds have been mainly found in species belonging to the Fabaceae family. Several plants that contain pterocarpans have been used in traditional medicine of many diverse cultures as an oral antidote against the venoms of snakes and spiders [17] and previous studies revealed that pterocarpan containing plants have antifungal, antibacterial, antitumoral and anti-inflammatory activities [18,19,20]. So far, potential antidepressant effects of some isoflavones have been evaluated and the involvement of noradrenergic, serotonergic and dopaminergic mechanisms have been thought to be responsible for the activity [16]. However, medicarpin derivative compounds have never been studied for their antidepressant effects [21,22].

Gossypetin is a hexahydroxylated flavonoid and named in chemistry as 3,5,7,8,3’,4’ hexahydroxy flavone. There are several studies that have shown that this compound has diuretic, antioxidant, antimicrobial, antimutagenic, antiatherosclerotic, cytoprotective, anxiolytic and antidepressant effects [23,24,25]. In a study that was designed to investigate the antidepressant and antianxiety potentials of *Hibiscus sabdariffa* Linn. (Malvaceae) calyces, which have been used traditionally as a sedative and for treating other nervous disorders, gossypetin has been shown to exhibit significant antidepressant and antianxiety activity at the dose of 20 and 5 mg/kg po, respectively [26] Gossypetin-8-*O*-β-d-glucuronide which isolated from *Abelmoschus manihot* (L.) Medic. has been showed an obvious antidepressant activity via up-regulation of BDNF expression [27]. 

Naringenin is a naturally occurring flavanone known to have anticancer, antimutagenic, anti-inflammatory, antioxidant, antiproliferative, hepatoprotective and antiatherogenic activities [28,29,30,31]. Naringenin was also studied for its antidepressant activity and it was observed that this compound possessed powerful anti-depressant like activity via the central serotonergic and noradrenergic systems [32,33]. 

Currently commercially available drugs for the treatment of depression are associated with various adverse effects and have many other disadvantages such as relatively low response and problematic interactions [34]. For these reasons, searches for novel pharmacotherapeutics from medicinal plants are still ongoing. 

## 4. Materials and Methods 

### 4.1. Plant Material

Fresh aerial parts of *L. corniculatus* subsp. *corniculatus* were collected on the date of 18.05.2018 from the roadside of Isparta-Konya highway in Turkey and authenticated by Prof. Dr. Hasan ÖZÇELİK from Süleyman Demirel University, Department of Biology, Faculty of Science and Art, Isparta. A voucher specimen with the code GUEF3467 was deposited in the Herbarium of the Faculty of Pharmacy, Gazi University, Ankara, Turkey.

### 4.2. Extraction, Fractionation and Isolation Process

Firstly, considering the traditional usage of the plant, milled dry aerial parts were extracted with distilled water at room temperature. As the aqueous extract did not show any remarkable activity in used experimental models of depression in mice, n-hexane, ethyl acetate (EtOAc) and methanol (MeOH) extracts were prepared sequentially through maceration method. The percentage yield of the EtOAc extract (12.4%) was higher than n-hexane (4.6%) and methanolic (11.7%) extracts. Since the MeOH extract of the plant material was found active in experiments, it was decided to fractionate. 

#### 4.2.1. Fractionation of MeOH Extract

The MeOH extract (16 g) was subjected to RP-18 column vacuum liquid chromatography (H_2_O (2 L), H_2_O:MeOH (90:10, 1 L), H_2_O:MeOH (80:20, 1 L), H_2_O:MeOH (70:30, 2 L), H_2_O:MeOH (60:40, 2 L), H_2_O:MeOH (50:50, 2 L), H_2_O:MeOH (40:60, 2 L), H_2_O:MeOH (30:70, 2 L), H_2_O:MeOH (20:80, 2 L), H_2_O:MeOH (10:90, 1 L), MeOH (1 L), acetone (1 L)) to obtain 42 fractions, which were combined as follows Fractions 1–42 after thin-layer chromatography (TLC) control (Mobile phase: EtOAc:CHCl_3_:MeOH:H_2_O (6:4:4:1)). Since the Fr. (13–20) fraction were displayed the potent antidepressant activity, this fraction applied to Sephadex LH-20 (25–100 μm, Sigma Chem. Co.) column using MeOH to obtain pure compounds.

#### 4.2.2. Determination of the Structure of Compounds

The structural determination of the compounds was achieved by mass spectroscopy (MS; Agilent G6550A Q-TOF mass spectrometer and Waters LCT Premier XE UPLC/MS-TOF spectrometer) and nuclear magnetic resonance (^1^H and ^13^C NMR; Bruker Ascend spectrometer) techniques. Totally 3 compounds were isolated from active fraction and the isolates were identified medicarpin-3-*O*-glucoside (a pterocarpan glycoside), gossypetin-3-*O*-glucoside (a flavonol glycoside) and naringenin-7-*O*-glucoside (prunin; a flavanone glycoside) by comparing their spectroscopic data with related literatures [35,36,37].

### 4.3. Biological Activity Tests

#### 4.3.1. Animals

The experiments were conducted on male BALB/c mice (25–30 g) and male Sprague-Dawley male rats (180–200 g), which were provided from Kobay Animals Laboratory (Ankara, Turkey) and kept in 12-h light/dark cycle for 3 days at room conditions with free access to laboratory food and water tap ad libitum. Six animals were used for each experimental group and all of the in vivo behavioral experiments were performed conferring to the international rules regarding the animal experiments and biodiversity rights (Kobay Ethical Council Project Number: 255).

#### 4.3.2. Preparation of Test Samples for Bioassay

All of the extracts were suspended in sodium carboxymethyl cellulose solution (0.5% CMC) using an ultrasonic bath and test samples were administered to the test group animals by oral gavage at the dose of 100 mg/kg. In the control group, animals received only 0.5% CMC solution to eliminate the effect of carrier material on biological activity and determine truly the effectiveness of the test materials, Fluoxetine (25 mg/kg dose; Merck, Darmstadt, Germany) and imipramine (30 and 50 mg/kg doses; Merck, Darmstadt, Germany) in 0.5% CMC were selected as reference drugs.

#### 4.3.3. Forced Swimming Test

Forced swimming test (FST) was carried out according to the methodology described by Porsolt et al. (1977) with some modifications [38,39]. This test model is based on that the inactivity of mice reflects the behavioral despair seen in depressive people. One hour after administering the test materials, the mice were individually forced to swim in a 20-cm-high transparent glass cylinder containing 8-cm-deep water at 21–24 °C and left there for 6 min. The water in the vessel was changed after each trial. Each animal was used only once. The test was videotaped, and the total duration of immobility was measured during the last 4-min interval of the test. Immobility, climbing and swimming behaviors were recorded. The mice were considered immobile only when they made no additional attempts to escape except the movements needed to keep their heads above the water. Six mice were used for each test group.

#### 4.3.4. Tail Suspension Test

Tail suspension test (TST) was performed as described by Steru et al. (1985) with minor modifications [40,41]. This test is based on the determination of the period of immobility of mice like FST. The mice, both acoustically and visually isolated, were hung 50 cm above the floor with the help of an adhesive tape placed approximately 1 cm from the tip of the tail one hour after administering the test samples orally. The test was videotaped, and the total duration of immobility time was chronometered for 6 min of the 10-min period. The mice were considered immobile only when they did not show any body movement, hung passively and stayed completely motionless for at least 1 min. Six mice were used for each test group.

#### 4.3.5. Antagonism of Hypothermia and Ptosis Induced by Tetrabenazine

The antagonism of tetrabenazine-induced hypothermia, ptosis and suppression of locomotor activity test was performed as described by Alpermann et al. [42]. This test is based on the observation of changes in the ptosis scores, rectal temperatures and durations of immobilization of mice. The mice with rectal temperatures of 36–38 °C were used in these experiments. One hour after oral administration of suitable materials for the animals in the control and experimental groups, tetrabenazine, which was dissolved in 0.1 M aqueous tartaric acid solution and adjusted to pH 6 with 10% NaOH administered intraperitoneally at 32 mg/kg. Thirty minutes after administration, the animals were placed one by one in the center of a 20 cm diameter disc and their locomotor activity percentages, ptosis scores and rectal temperature changes were determined. Mice walking to the edge of the disc and looking to the edge, rotating 180° in place, or moving the head 90° in one direction were not considered akinetic. The following formula was used to calculate the percentage of animals showing locomotor activity:

Locomotor activity (%) = (Number of non-akinetic mice × 100)/Number of mice in group

The ptosis scoring was done according to the following scale:

0: Eyes open;

1: One quarter off;

2: Half closed;

3: Three quarters off;

4: Fully enclosed.

Sixty min after the administration of tetrabenazine, the rectal temperature of each mouse was measured again with a clinical digital thermometer (BIO-TK9882 Rodent Thermometer) to record the change. Six mice were used for each test group.

#### 4.3.6. The Inhibitory Activity on the MAO A&B

The effects of prepared extracts and fractions obtained from MeOH extract on the activity of MAO-A and B enzymes were investigated in vitro in the brain and liver of the rats according to the methods reported by Hwang et al. (2003) and Küpeli Akkol et al. (2019) with some modification [43,44]. The anesthetized rats were lost blood with 3.13% sodium citrated syringe from heart. The brain tissue was obtained from the decapitated brain, which was washed with 0.01 M phosphate buffered saline (PBS, pH 7.0), and homogenate at 40 °C for 1 min followed by added cold 0.25 M sucrose by 9 parts of wet weight of tissue. Centrifuged at 700× *g* in 40 °C for 20 min. Supernatant was centrifuged at 18,000× *g* for 20 min immediately. Pellet was suspended in 5 parts of PBS, and used for crude enzyme preparation. Prepared crude MAO-A (0.5 mL) was added to test tubes with 1.0 mL of test materials. It was incubated in shaking incubator at 37.5 °C for 15 min. As a substrate, 0.5 mL of 1.0 mM serotonin was added and incubated at 37.5 °C for 90 min. To terminate the enzyme action, test tubes were heated at 95 °C water bath for 3 min. and centrifuged at 700× *g* for 20 min. immediately. Supernatants were poured in prepared Amberlite CG-50 (H+ form) column (0.6 cm × 4 cm). After washed with distilled water thoroughly (over 40 mL), eluted with 3 mL of 4 N acetic acid, elute was determined of absorbance at 277 nm. Instead of samples, same volumes of distilled water were added in control. In the sample controls, the substrates were added on the time of activity termination instead of initiation of action. Each group was performed with duplicated and calculated for the inhibition percentages of samples by proper expression.

## 5. Conclusions 

The data of this study demonstrated that the MeOH extract of the aerial parts of *L. corniculatus* subsp. *corniculatus* showed a remarkable in vivo antidepressant effect and the flavonoids isolated from the active subfractions could be responsible from the activity. Further studies are planned to undertaken to determine what is the mechanism of these compounds to develop new antidepressant-acting drugs.

## Figures and Tables

**Figure 1 molecules-25-01299-f001:**
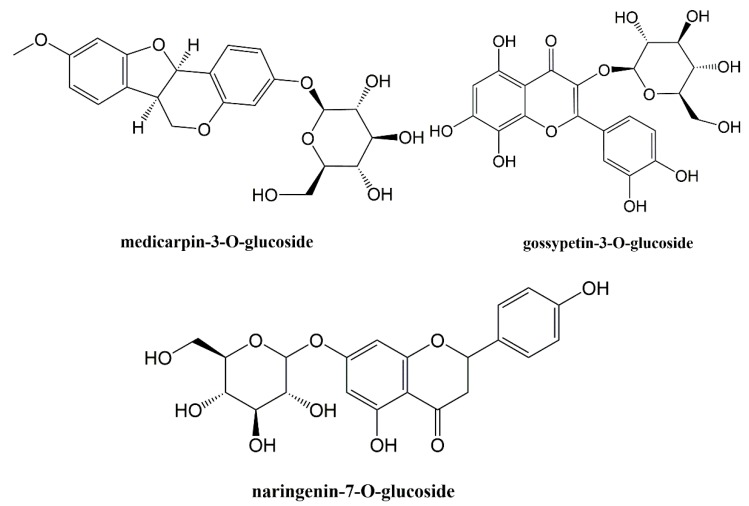
Chemical structures of isolated compounds.

**Table 1 molecules-25-01299-t001:** Effects of the aqueous extracts prepared from the aerial parts of *Lotus corniculatus* L. subsp. *corniculatus* in the antidepressant activity tests.

**Forced Swimming Test**				
**Material**	**Dose (mg/kg. p.o.)**	**Duration of Immobility (s) (Mean ± S.E.M.)**	**Variation (%)**	
Control	-	205.13 ± 22.54	-	
Aqueous extract	100	204.66 ± 22.49	−0.23	
Imipramine HCl	30	**102.87 ± 9.98 ****	**−49.85**	
	50	**85.41 ± 7.64 *****	**−58.36**	
**Tail Suspension Test**				
Control	-	215.37 ± 27.01	-	
Aqueous extract	100	204.60 ± 22.40	−5.00	
Imipramine HCl	30	**82.81 ± 7.82 *****	**−61.55**	
	50	**71.26 ± 6.91 *****	**−66.91**	
**Antagonism of Tetrabenazine-Induced Ptosis, Hypothermia and Suppression of Locomotor Activity**				
**Material**	**Dose (mg/kg)**	**Ptosis Mean Score (Mean ±S.E.M.)**	**Locomotor Activity (%)**	**Mean Decrease in Rectal Temperature (°C) (Mean ± S.E.M.)**
Control	-	3.83 ± 1.27	0.00	5.12 ± 0.43
Aqueous extract	100	3.50 ± 1.19	0.00	4.22 ± 0.37
Fluoxetine HCl	25	**0.00 ± 0.00 *****	**100.00 *****	**0.30 ± 0.03 *****

**: *p* < 0.01; ***: *p* < 0.001 (S.E.M.: Standard Error of the Mean).

**Table 2 molecules-25-01299-t002:** Effects of the extracts and fractions prepared with organic solvents from *Lotus corniculatus* L. subsp. *corniculatus* in the forced swimming test.

Effects of the Extracts			
Material	Dose (mg/kg)	Duration of Immobility (s) (Mean ± S.E.M.)	Variation (%)
Control	-	210.17 ± 25.59	-
*n*-Hexane extract	100	142.31 ± 20.86	−32.29
EtOAc extract		165.40 ± 25.02	−21.30
**MeOH extract**		**139.97 ± 19.03 ** **	**−33.40**
Imipramine HCl	30	**115.76 ± 10.48 ****	**−44.92**
	50	**91.27 ± 8.63 *****	**−56.57**
**Effects of the Fractions Obtained from Active MeOH Extract**			
Control	-	205.21 ± 22.31	-
Fr. (1−12)	100	129.42 ± 13.09	−36.93
**Fr. (13–20)**		**115.22 ± 11.03 ****	**−43.85**
Fr. (21–35)		140.24 ± 18.03	−31.66
Fr. (36–42)		150.46 ± 19.02	−26.68
Imipramine HCl	30	**106.11 ± 9.85 ****	**−48.29**
	50	**85.47 ± 7.40 *****	**−58.35**

**: *p* < 0.01; ***: *p* < 0.001 (S.E.M.: Standard Error of the Mean).

**Table 3 molecules-25-01299-t003:** Effects of the extracts and fractions prepared with organic solvents from *Lotus corniculatus* L. subsp. *corniculatus* in the tail suspension test.

**Effects of the Extracts**			
**Material**	**Dose (mg/kg)**	**Duration of immobility (s) (Mean ± S.E.M.)**	**Variation (%)**
Control	-	203.50 ± 22.13	-
*n*-Hexane extract	100	131.99 ± 17.03	−35.14
EtOAc extract		140.38 ± 19.87	−31.02
**MeOH extract**		**125.94 ± 16.57 ****	**−38.11**
Imipramine HCl	30	**88.11 ± 8.01 *****	**−56.70**
	50	**75.37 ± 7.32 *****	**−62.96**
**Effects of the Fractions Obtained from Active MeOH Extract**			
**Material**	**Dose (mg/kg. p.o.)**	**Duration of immobility (s) (Mean ± S.E.M.)**	**Variation (%)**
Control	-	210.11 ± 25.05	-
Fr. (1−12)	100	158.27 ± 19.29	−24.67
**Fr. (13–20)**		**115.41 ± 14.17 ****	**−45.07**
Fr. (21–35)		145.36 ± 18.46	−30.82
Fr. (36–42)		131.12 ± 19.73	−37.59
Imipramine HCl	30	**90.02 ± 9.71 ****	**−57.16**
	50	**81.46 ± 6.42 *****	**−61.23**

**: *p* < 0.01; ***: *p* < 0.001 (S.E.M.: Standard Error of the Mean).

**Table 4 molecules-25-01299-t004:** Effects of the extracts and fractions prepared with organic solvents from *Lotus corniculatus* L. subsp. *corniculatus* in the antagonism of tetrabenazine-induced ptosis, hypothermia and suppression of locomotor activity tests.

Effects of the Extracts				
Material	Dose(mg/kg)	Ptosis mean score(Mean ± S.E.M.)	Locomotor Activity (%)	Mean Decrease in Rectal Temperature (°C; Mean ± S.E.M.)
Control	-	3.83 ± 1.29	0.00	5.01 ± 0.43
*n*-Hexane extract	100	2.25 ± 0.86	16.7	1.46 ± 0.11
EtOAc extract		2.50 ± 0.96	33.3	3.28 ± 0.27
**MeOH extract**		**1.75 ± 0.52 ** **	**66.7 ****	**1.51 ± 0.13 ****
Fluoxetine HCl	25	**0.00 ± 0.00 *****	**100.00 *****	**0.24 ± 0.02 *****
**Effects of the Fractions Obtained from Active MeOH Extract**				
Control	-	3.82 ± 1.23	0.00	5.23 ± 0.47
Fr. (1–12)	100	2.23 ± 0.82	16.7	3.28 ± 0.27
**Fr. (13–20)**		**0.15 ± 0.03 ****	**66.7 ****	**0.58 ± 0.07 ****
Fr. (21–35)		2.27 ± 0.83	16.7	3.82 ± 0.33
Fr. (36–42)		2.11 ± 0.76	33.3	2.68 ± 0.18
Fluoxetine HCl	25	**0.00 ± 0.00 *****	**100.00 *****	**0.27 ± 0.02 *****

**: *p* < 0,01; ***: *p* < 0,001 (S.E.M.: Standard Error of the Mean).

**Table 5 molecules-25-01299-t005:** Effect of extracts and fractions obtained from *Lotus corniculatus* L. subsp. *corniculatus* on the MAO inhibition assay.

Material	IC_50_ (mg/mL) ± S.D.	
	MAO-A	MAO-B
Control	0.182 ± 0.022	0.115 ± 0.046
Aqueous extract	3.273 ± 0.149	2.371 ± 0.058
n-Hexane extract	2.154 ± 0.927	1.196 ± 0.136
EtOAc extract	4.649 ± 1.023	5.182 ± 1.049
MeOH extract	9.315 ± 2.719	7.109 ± 1.172
Fr. (1–12)	1.408 ± 0.612	3.524 ± 1.281
Fr. (13–20)	8.923 ± 2.041	9.305 ± 2.972
Fr. (21–35)	3.287 ± 1.925	1.471 ± 0.228
Fr. (36–42)	1.054 ± 0.258	2.106 ± 0.130
	**IC_50_ (M)**	
Caffeine	0.004 ± 0.001	0.003 ± 0.001

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
