# Peer review of "Antidepressant Potential of *Lotus corniculatus* L. subsp. *corniculatus*: An Ethnobotany Based Approach"

_molecules, 2020, doi:10.3390/molecules25061299_

Round 1
Reviewer 1 Report
This manuscript deals with the potential Antidepressant action showed by Lotus corniculatus. Several pharmacological test have been explored and the isolation of three main metabolites is reported. The manuscript is adequate for Molecules, pending the following changes:
Structures of figure 1 must be improved
Three compounds were isolated using Sephadex LH-20 with MeOH but, how were the metabolites identified/elucidated?
Authors have isolated 3 main compounds, but there are not enough evidence that they are the responsible of the antidepressant action of Lotus corniculatus. Comment about it.
Authors remarked that there is the first report about the antidepressant action of this plant. However several flavonoids have demonstrated CNS action, such as tilianin with anxiolytic-like activity (see Gonzalez.Trujano et al. Asian Pacific Journal of Tropical Medicine 2015, 185-190). Please include this info in the proper section.
IC50 must be written in subscript
Conclusion is missing in the manuscript
Author Response
Structures of figure 1 must be improved
Structures of figure 1 was improved
Three compounds were isolated using Sephadex LH-20 with MeOH but, how were the metabolites identified/elucidated?
In Section 5 (Material and Methods) is mentioned as;
5.2.1. Fractionation of MeOH Extract
The MeOH extract (16 g) was subjected to RP-18 column vacuum liquid chromatography [H2O (2 L), H2O:MeOH (90:10, 1 L), H2O:MeOH (80:20, 1 L), H2O:MeOH (70:30, 2 L), H2O:MeOH (60:40, 2 L), H2O:MeOH (50:50, 2 L), H2O:MeOH (40:60, 2 L), H2O:MeOH (30:70, 2 L), H2O:MeOH (20:80, 2 L), H2O:MeOH (10:90, 1 L), MeOH (1 L), acetone (1 L)] to obtain 42 fractions which were combined as follows Fractions A–D after thin-layer chromatography (TLC) control [Mobile phase: EtOAc:CHCl3:MeOH:H2O (6:4:4:1)]. Since the fraction B were displayed the potent antidepressant activity, this fraction applied to Sephadex LH-20 (25-100 mm, Sigma Chem. Co.) column using MeOH to obtain pure compounds.
5.2.2. Determination of the Structure of Compounds
The structural determination of the compounds was achieved by mass spectroscopy (MS; Agilent G6550A Q-TOF mass spectrometer and Waters LCT Premier XE UPLC/MS-TOF spectrometer) and nuclear magnetic resonance (1H and 13C NMR; Bruker Ascend spectrometer) techniques. Totally 3 compounds were isolated from active fraction and the isolates were identified medicarpin-3-O-glucoside (a pterocarpan glycoside), gossypetin-3-O-glucoside (a flavonol glycoside) and naringenin-7-O-glucoside (prunin) (a flavanone glycoside) by comparing their spectroscopic data with related literatures [35-37].
Authors have isolated 3 main compounds, but there are not enough evidence that they are the responsible of the antidepressant action of Lotus corniculatus. Comment about it.
In this study, isolation of the compound/s responsible for the effect of MeOH extract, which is active in biological activity experiments by chromatographic methods, is aimed. In our study, bioassay guided active fractionation and isolation technique was performed on MeOH extract. Here, the compounds of the active extract are isolated and their structures are elucidated. However, since the amounts of the compounds could not be obtained enough amount to be tested in biological activity experiments, the effects of these compounds were discussed by attributing to the previous activity studies.
Authors remarked that there is the first report about the antidepressant action of this plant. However several flavonoids have demonstrated CNS action, such as tilianin with anxiolytic-like activity (see Gonzalez.Trujano et al. Asian Pacific Journal of Tropical Medicine 2015, 185-190). Please include this info in the proper section.
This information is completely correct. This is the first report to evaluate the antidepressant activity of the Lotus corniculatus. The article shown by the referee as an example belongs to the Agastache mexicana plant and has nothing to do with the plant we studied on.
IC50 must be written in subscript
IC50 was written in subscript
Conclusion is missing in the manuscript
Coclusion was added.

Reviewer 2 Report
Dareli et al. investigated the antidepressant properties of different extracts of the aerial parts of a typical Turkish plant Lotus corniculatus.
I particularly appreciated the experimental design of this study and its objectiveness in linking results and discussion.
However, there are still some pending questions to be answered before its acceptance. Point-by-point, these questions/comments are shown below:
(1) From my point of view, whenever mentioning ethnobotanical specimens from different countries, an illustrative picture of it is almost mandatory. The whole plant from a distance (in natural habitat) and a close-up of the most relevant part applied in the study (the aerial parts here)
(2) Authors have to provide more information about the freshness, overall health, quality, and eventual presence of contaminants (heavy metals, polyaromatic hydrocarbons, and other pollutants, etc.) of the collected specimens, since the plants were originally collected from roadside a highway.
(3) The authors have to justify why they used such a high dose as 100 mg/kg.
(4) Please, all method descriptions need more details. Specify all "modifications on methods".
(5) Please, reformat Table 1 to avoid "Imipramine.HCl" split in two lines. As it stands, the table gives the (absurd) wrong idea that, somehow, the study applied 50 mg/kg HCl o.p. in the animals.
(6) Please, check the number of digits of "Tail suspension (s)" and "Duration of immobility (s)" values (± SD). For example, (206.66 ± 22.49) s shouldn't be (206.7 ± 22.5) or did you actually have an accurate instrument with that precision?
(7) [Line 93] Please, improve the description of IC50 MAO inhibition results. Moreover, present numeric values (±SD) of all extracts and fractions shown in Table 5. This is a very important biochemical evidence of your study.
(8) Please, provide LC/MS mass chromatograms (at least) to illustrate main peaks characterization of the three main activie compounds of MeOH extracts.
Author Response
- From my point of view, whenever mentioning ethnobotanical specimens from different countries, an illustrative picture of it is almost mandatory. The whole plant from a distance (in natural habitat) and a close-up of the most relevant part applied in the study (the aerial parts here)
The photograph of the plant's image in nature and the herbarium sample are given in the graphical abstract.
- Authors have to provide more information about the freshness, overall health, quality, and eventual presence of contaminants (heavy metals, polyaromatic hydrocarbons, and other pollutants, etc.) of the collected specimens, since the plants were originally collected from roadside a highway.
Many thanks for the reviewer’s suggestion. Information about the overall health, quality, and eventual presence of contaminants (heavy metals, polyaromatic hydrocarbons, and other pollutants, etc.) of the collected specimens are required for a study in the toxicolgical field. Since the mentioned studies are outside our field, these experiments have not been included.
- The authors have to justify why they used such a high dose as 100 mg/kg.
If researchers don’t have any information about the aplication dosage, they usually select the dose of 100 mg/kg for test samples. So, this dose was not found high.
- Please, all method descriptions need more details. Specify all "modifications on methods".
Done
- Please, reformat Table 1 to avoid "Imipramine.HCl" split in two lines. As it stands, the table gives the (absurd) wrong idea that, somehow, the study applied 50 mg/kg HCl o.p. in the animals.
Done
- Please, check the number of digits of "Tail suspension (s)" and "Duration of immobility (s)" values (± SD). For example, (206.66 ± 22.49) s shouldn't be (206.7 ± 22.5) or did you actually have an accurate instrument with that precision?
Done
- [Line 93] Please, improve the description of IC50 MAO inhibition results. Moreover, present numeric values (±SD) of all extracts and fractions shown in Table 5. This is a very important biochemical evidence of your study.
Done
- Please, provide LC/MS mass chromatograms (at least) to illustrate main peaks characterization of the three main activie compounds of MeOH extracts.
LC / MS technique was not used in this study. Directly isolation was done. NMR and MS information was not given since the isolated substances were known. This information will be provided if requested.
